# VISION-LANGUAGE SUBSPACE PROMPTING

## ABSTRACT

Prompting vision-language models like CLIP to adapt to downstream tasks is currently topical. A seminal technique to this end is context optimization, which replaces a subset of textual tokens with trainable parameters (a.k.a *soft prompts*). However, current pipelines use a single vector embedding induced by soft prompts as the classifier weight for visual recognition. This can lead to problems where the learned soft prompts overfit to base classes' training data, resulting in poor performance when applied to novel classes. Several approaches were proposed to address this issue by regularizing the learned soft prompts to align them with hand-crafted text/hard prompts. However, excessive regularization of the soft prompts can hurt the model's performance on the base classes it is trained on. Maintaining the right balance to ensure strong base- and novel-class performance is crucial but non-trivial. In this paper, we introduce a novel subspace-based prompt learning method, named SuPr, which can effectively model subspaces spanning the embeddings of both the learnable soft and the textual/hard prompts. Our subspace-based alignment between hand-crafted and learnable prompts balances these effects to achieve excellent fitting of base classes as well as generalization to novel classes. With the advantages of subspace modelling, our SuPr shows its effectiveness on generalization from base to new, domain generalization, cross-dataset transfer and few-shot learning, leading to new state-of-the-art results in all settings.

## 1 INTRODUCTION

The recent advances in large-scale Visual-Language Models (VLMs), exemplified by models such as CLIP (Radford et al., 2021) and ALIGN (Jia et al., 2021), have attracted substantial attention. These VLMs align visual and textual modalities thanks to their extensive pretraining on large sets of visual and textual data pairs. The learned common feature space between two modalities grants VLMs exceptional performance across various zero-shot tasks (Alayrac et al., 2022). However, despite their great zero-shot performance, maximizing VLM performance on a downstream task explicitly is naturally desirable (Gao et al., 2023).

One pervasive strategy is *prompt engineering* (Radford et al., 2021), which optimizes the textual prompt while keeping other VLMs parameters fixed. For one example of using VLMs for image recognition, when these optimized prompt sentences are coupled with class names as input to the text encoder, the pre-trained VLMs achieve an enhanced class-specific classifier. Specifically, by transforming prompts such as "`a photo of a {CLASS}.`" into "`a photo of a {CLASS}, a type of bird.`", the model effectively transitions from a generic model of object classification to a specific one of bird classification. This approach helps adapt VLMs to diverse tasks; however, its optimization relies on handcrafting. Therefore, *prompt learning* (Zhou et al., 2022b) as an alternative has emerged, which replaces part of discrete textual/hard tokens with continuously learnable parameters, a.k.a "soft prompts", that can be updated through end-to-end training. Soft prompt learning has proven effective by outperforming handcrafted prompt optimization when they are tailored for known (base) classes (Yu et al., 2023; Lu et al., 2022). However, it introduces a new problem — base-class overfitting. Essentially, the learned soft prompts overfit to solving tasks of the base classes, and the performance significantly declines when applied to recognize novel classes never encountered at training (Zhou et al., 2022a). This is in contrast with optimizing handcrafted prompts, which do not break a VLM's zero-shot capacity for novel classes.

The main current challenge in prompt learning is maintaining a VLM's generalizable performance on novel classes while tailoring them for the known base classes. To this end, most existing works

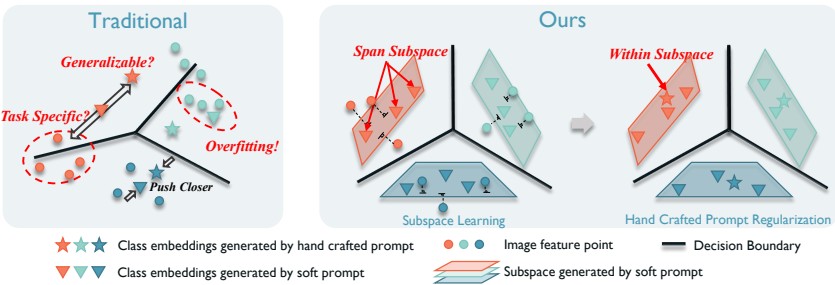

Figure 1: A schematic illustration of our SuPr.

focus on aligning (handcrafted) hard[1] and soft prompts (Bulat & Tzimiropoulos, 2023; Yao et al., 2023; Zhu et al., 2023), such that the learned soft prompts still retain zero-shot generalization from pre-training. Despite being effective, such a regularization restricts the extent of optimizing soft prompts, thus potentially hurting their performance on base classes, as shown in some experimental evaluations. In this paper, we take a different stance — we no longer impose strict alignment between soft- and hard-prompt embeddings and instead propose *subspace prompting* (SuPr) to model the vector subspaces that can accommodate both the soft- and hard-prompt embeddings jointly, as illustrated in Figure 1.

Specifically, instead of training a single set of soft prompts to fit the base classes, we introduce multiple sets of soft prompts as the trainable parameters. We divide the single set of soft prompts proposed in CoOp (Zhou et al., 2022b) into multiple sets with shorter lengths, maintaining the same amount of total trainable parameters. Then, we generate multiple token sequences using different sets of soft prompts for the same class and feed each of them into the text encoder to achieve an embedding. Having a set of text embeddings representing the same class, we can form a subspace considering those embeddings as support points. Given an image embedding and a text embedding, CLIP calculates the cosine similarity between two embeddings for predictions. In contrast, we can calculate the projection point of this image embedding on the text subspace and use the point-to-subspace distance — namely, the distance between the original and projection embeddings — as an alternative. Classification of an image sample is now equivalent to finding the nearest text subspace by comparing point-to-subspace distances.

Now, in SuPr, the text embeddings induced by different sets of soft prompts can be relaxed to capture different underlying aspects of the visual embeddings from the same class. Unlike a conventional classifier using a single-point vector as the weights, classifiers using subspaces have the extrapolation capability to make predictions/estimates beyond the observed data points, thus having better generalization (Simon et al., 2020). Despite better generalization, classifiers with trained subspaces may still break the embedded zero-shot generalization of a pre-trained VLM after finetuning on a downstream task. However, now we don't force the text embeddings induced by soft prompts to align with those induced by hard prompts. Instead, we can tune the modelled subspace to span the hard-prompt induced text embeddings to retain the zero-shot generalization for novel classes. Our method does not restrict the number of hard prompts used for regularization. However, when various types of hard prompts are proposed, the difficulty of spanning all of them in the same subspace may be introduced. We empirically discover that modelling multiple subspaces and using different groups of hard prompts with similar constraints to regularize different individual subspaces results in a strong ensembling.

We have conducted a series of rigorous experiments across various datasets, spanning the settings of generalization from base to new, domain generalization, cross-dataset transfer and few-shot learning. The substantial improvements achieved in these experiments underscore the effectiveness of our approach in learning representative prompts for subspace modelling.

## 2 RELATED WORK

**Vision-Language Models** Vision-Language Models (VLMs) represent a cutting-edge fusion of computer vision and natural language processing, bridging the gap between visual and textual data.

---

[1] We sometimes refer to the handcrafted prompts as hard prompts.

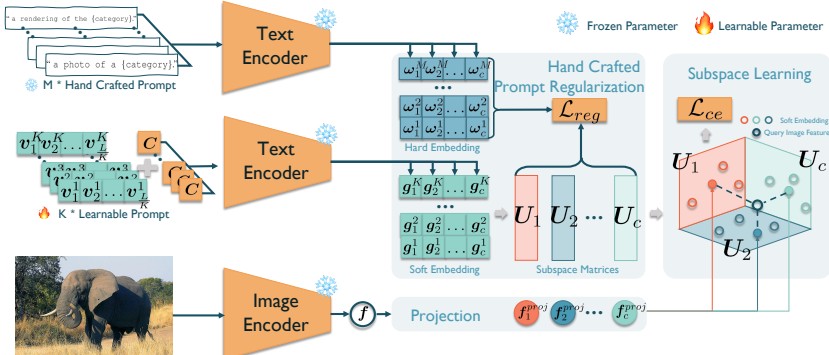

Figure 2: The architecture of our SuPr.

In recent paradigms, VLMs are pre-trained using vast image-text pairs from the internet (Schuhmann et al., 2021; 2022), enabling them to understand image-text relationships, form a foundational understanding of the visual world, and excel in open-world scenarios (Radford et al., 2021; Jia et al., 2021). VLMs can be categorized into three types based on their objective functions: contrastive (Mu et al., 2022; Li et al., 2022a), generative (He et al., 2022), and alignment objectives (Singh et al., 2022; Li et al., 2022b).Additionally, researchers are actively exploring more parameter-efficient fine-tuning methods for VLMs. Techniques such as prompt learning and adaptor-based Gao et al. (2023); Zhang et al. (2022) approaches have attracted substantial interest, holding potential for tailoring VLMs to specific downstream tasks. This paper builds upon the prompt learning paradigm and introduces a novel prompt-tuning approach based on subspace modelling.

**Prompt Learning** Prompt learning (Jiang et al., 2020; Shin et al., 2020; Li & Liang, 2021) has emerged as a powerful method for adapting VLMs to a wide array of tasks. CoOp (Zhou et al., 2022b) pioneered the integration of soft prompts into VLMs, transitioning prompts from discrete word tokens to continuous, trainable parameters. This pivotal innovation has fueled the rapid evolution of prompt learning within the VLM domain. However, the potential of soft prompt learning faces the challenge of base class overfitting. To address this concern, CoCoOp (Zhou et al., 2022a) improves the generalization of soft prompts by introducing instance-conditional prompt refinement. KgCoOp (Yao et al., 2023) tackles the issue through regularization, constraining the distance between hard and soft prompts. Similarly, LASP (Bulat & Tzimiropoulos, 2023) incorporates a text-to-text loss to confine the optimization of soft prompts within a language-aware embedding space. ProGrad (Zhu et al., 2023) fine-tunes the gradient by projecting it, ensuring that it stays aligned with the textual prompt direction throughout the optimization process. Additionally, some studies focus addressing these challenges through a richer prompt representation. Notably, ProDA (Lu et al., 2022) models prompts as mixtures of Gaussian distributions, while PLOT (Chen et al., 2022) represents prompts as multiple attributes and introduces optimal transport metrics to measure their similarity. In the sprit of these works, we model prompts as subspaces, which have the capacity to encompass a denser spectrum of visual concepts. This approach strikes a superior balance between training performance and generalization capability, effectively alleviating base class overfitting.

**Subspace Modelling** Subspace modelling is a versatile technique used in modelling concepts with intra-class variability, such as diverse attributes. Instead of relying on a single-point estimate, learning a subspace spanning the variability of a class is a more effective choice. Subspaces have found wide-ranging applications in fields such as few shot learning (Simon et al., 2020; Cheraghian et al., 2021; Zhu & Koniusz, 2022; Gao et al., 2021), face recognition (Wang & Tang, 2004; Wang & Tang), and clustering (Ji et al., 2017; Zhou et al., 2018; Cai et al., 2022). DSN (Simon et al., 2020) introduced subspace classifiers for the first time in few-shot learning, showcasing their superior data-efficient modelling capabilities compared to prototype networks. This highlights the potential of subspace classifiers in prompt learning. Wang & Tang (2004) proposed constructing a set of faces within a subspace, including distinctions in various facial attributes. This demonstrates the ability of subspaces to represent rich semantics. It's worth noting that SubPT (Ma et al., 2023), a closely related concurrent work, uses gradient subspaces to constrain optimization to prevent a model from learning spurious correlations on the training data. This approach diverges significantly from our strategy of modelling subspaces using soft prompts, highlighting a distinct conceptual foundation. To the best of our knowledge, we are the first to integrate soft-prompt subspace modeling into VLMs.

# 3 APPROACH

## 3.1 PRELIMINARIES

Our method applies to different existing VLMs. For simplicity and fair comparisons to the prior works in this stream, we stick with the Contrastive Language-Image Pre-training (CLIP) model (Radford et al., 2021). Our overall framework is illustrated in Figure 2, and we provide further details in the following sections.

**CLIP/Prompt Engineering** When performing a $C$-way image classification using CLIP, each class is associated with a specific sentence, denoted as $\boldsymbol{t}_i$, $i \in \{1, \ldots, C\}$. These sentences follow some common *prompt* templates, for instance, "a photo of a {CLASS}." where the "{CLASS}" placeholder can be filled with actual class names. These word tokens are then fed into CLIP's text encoder $g(\cdot)$ to generate text embeddings, represented as $\boldsymbol{\omega} = \{ \boldsymbol{\omega}_c \mid \boldsymbol{\omega}_c = g(\boldsymbol{t}_c) \}_{c=1}^{C}$ where each $\boldsymbol{\omega}_c \in \mathbb{R}^{D \times 1}$. These text embeddings serve as the classifier weights for each class for recognizing an input image. Given an image feature $\boldsymbol{f} \in \mathbb{R}^{D \times 1}$ generated by the image encoder $f(\boldsymbol{x})$ for input image $\boldsymbol{x}$, the predictive probability for a particular class $c$ is determined as follows

$$p(c|\boldsymbol{f}) = \frac{\exp\left(\text{sim}\left(\boldsymbol{f}, \boldsymbol{\omega}_c\right)/\tau\right)}{\sum_{i=1}^{C} \exp\left(\text{sim}\left(\boldsymbol{f}, \boldsymbol{\omega}_i\right)/\tau\right)}, \tag{1}$$

where $\text{sim}(\cdot, \cdot)$ represents cosine similarity, and $\tau$ is a learned temperature parameter.

**CoOp/Prompt Learning** Unlike using discrete handcrafted hard prompts in prompt engineering, context optimization (CoOp) Zhou et al. (2022b) employs *soft prompts*, represented as continuous vectors. Then, these soft prompts can be optimized end-to-end for downstream tasks. Instead of using hard prompts like "a photo of a {CLASS}", CoOp replaces them with $\boldsymbol{T}_c = \{\boldsymbol{v}_1, \boldsymbol{v}_2, \ldots, \boldsymbol{v}_L, \boldsymbol{C}_c\}_{c=1}^{C}$, where $\{\boldsymbol{v}_1, \boldsymbol{v}_2, \ldots, \boldsymbol{v}_L\}$ represents $L$ learnable vectors and $\boldsymbol{C}_c$ is the token embedding for the class name. Analogously, classifier weights are derived from the soft prompts $\boldsymbol{g}_c = g(\boldsymbol{T}_c) \in \mathbb{R}^{D \times 1}$, and the predictive probability for class $c$ is given by

$$p(c|\boldsymbol{f}) = \frac{\exp\left(\text{sim}\left(\boldsymbol{f}, \boldsymbol{g}_c\right)/\tau\right)}{\sum_{i=1}^{C} \exp\left(\text{sim}\left(\boldsymbol{f}, \boldsymbol{g}_i\right)/\tau\right)}. \tag{2}$$

The learnable prompts can be optimized using cross-entropy loss $L_{ce}$

$$\{\boldsymbol{v}_1, \boldsymbol{v}_2, \ldots, \boldsymbol{v}_L\} := \underset{\{\boldsymbol{v}_1, \boldsymbol{v}_2, \ldots, \boldsymbol{v}_L\}}{\arg\min} -\sum_{c=1}^{C} \mathbf{1}[c == y] \cdot \log p(c|\boldsymbol{f}). \tag{3}$$

Here, $\mathbf{1}[.]$ is an indicator function. Note that, during the optimization, the pretrained image and text encoders $f(.)$ and $g(.)$ of CLIP remain frozen.

## 3.2 SUBSPACE MODELLING USING SOFT PROMPTS

In this section, we present a method for using soft prompts to model vector subspaces, and to subsequently employ the subspaces as classifiers for image classification.

Initially, CoOp introduced a single set of soft prompts, denoted as $\{\boldsymbol{v}_1, \boldsymbol{v}_2, \ldots, \boldsymbol{v}_L\}$, which results in a single text embedding serving as the classifier's weights. However, to mitigate the sub-optimal solutions discussed earlier, we adopt an approach to distribute this single set of soft prompts, denoted as $\boldsymbol{T} = \{\boldsymbol{v}_1, \boldsymbol{v}_2, \ldots, \boldsymbol{v}_L, \boldsymbol{C}\}$, into $K$ partitions, forming $\{\boldsymbol{T}^j\}_{j=1}^{K} = \{\boldsymbol{v}_1^j, \boldsymbol{v}_2^j, \ldots, \boldsymbol{v}_{\frac{L}{K}}^j, \boldsymbol{C}\}_{j=1}^{K}$ [2]. This distribution allows us to estimate $K$ distinct text embeddings, denoted as $\{\boldsymbol{g}^j = g(\boldsymbol{T}^j)\}_{j=1}^{K}$, all associated with the same class $\boldsymbol{C}$. In the following sections, we will elaborate on the process of modeling a vector subspace with $\{\boldsymbol{g}^j\}_{j=1}^{K}$ serving as support vectors.

Using the support vectors generated as described above, they are concatenated to create a matrix $\boldsymbol{S}_c \in \mathbb{R}^{D \times K}$. Then, we need to generate the linearly independent bases to form a subspace matrix. To this end, we employ singular value decomposition (SVD) of $\boldsymbol{S}_c$. To mitigate the potential issue of linear dependence among columns of $\boldsymbol{S}_c$, we introduce a relatively small Gaussian noise

---

[2]Importantly, this partitioning maintains the same total trainable parameters.

$\epsilon \in \mathbb{R}^{D \times K}$ into the $\boldsymbol{S}_c$ before SVD decomposition: $\boldsymbol{U}_c, \boldsymbol{\Sigma}_c, \boldsymbol{V}_c = \text{SVD}(\boldsymbol{S}_c + \epsilon)$. Then, we use orthonormal bases of the left singular matrix $\boldsymbol{U}_c$ for modelling the subspace of class $c$ denoted as $\boldsymbol{U}_c = [\boldsymbol{u}_1, \ldots, \boldsymbol{u}_K] \in \mathbb{R}^{D \times K}$

Given a subspace $\boldsymbol{U}_c$ for class $c$ and an image feature point $\boldsymbol{f}$, we replace $\text{sim}(\cdot, \cdot)$ the function introduced in Eq. 2, as follows

$$\text{sim}(\boldsymbol{f}, \boldsymbol{U}_c) = \text{cos\_sim}(\boldsymbol{f}, \boldsymbol{f}_c^{proj}). \tag{4}$$

Here, $\text{cos\_sim}(\boldsymbol{f}, \boldsymbol{f}_c^{proj})^3$ calculates the cosine similarity between the vector $\boldsymbol{f}$ and its projection $\boldsymbol{f}_c^{proj}$ onto the subspace $\boldsymbol{U}_c$. To find $\boldsymbol{f}_c^{proj}$ is to find the closest point $\boldsymbol{f}_c^{proj}$ to $\boldsymbol{f}_c$ in $\boldsymbol{U}_c$ by solving $\boldsymbol{\alpha} = \arg\min_{\boldsymbol{\alpha}} \|\boldsymbol{f} - \boldsymbol{U}_c \boldsymbol{\alpha}\|_2$ through a closed-form solution (Hastie et al., 2009). And given $\boldsymbol{U}_c$ is an orthonormal matrix, the projection can be formulated as follows:

$$\boldsymbol{f}_c^{proj} = \boldsymbol{U}_c \boldsymbol{U}_c^T \boldsymbol{f}, \qquad \boldsymbol{\alpha} = \boldsymbol{U}_c^T \boldsymbol{f}. \tag{5}$$

As a result, for an image feature $\boldsymbol{f}$, its predictive probability of belonging to class $c$ is calculated as

$$p(c|\boldsymbol{f}) = \frac{\exp\left(\text{sim}(\boldsymbol{f}, \boldsymbol{U}_c)/\tau\right)}{\sum_{i=1}^{K} \exp\left(\text{sim}(\boldsymbol{f}, \boldsymbol{U}_i)/\tau\right)}. \tag{6}$$

Then, we can optimize the trainable soft prompts similar to that in Eq. 3. As we discussed above, optimizing the modelled subspaces may break the zero-shot inference capacity of a pretrained CLIP for novel classes; thus, we introduce some regularization further to mitigate this problem.

### 3.3 REGULARIZATION USING HARD PROMPTS

To boost the generalization of learned soft prompts, both Yao et al. (2023) and Bulat & Tzimiropoulos (2023) have substantiated the effectiveness of maintaining a certain degree of alignment between soft prompts and handcrafted textual/hard prompts. Following this spirit, we regularize our modelled subspace of soft prompts $\boldsymbol{U}_c$ by forcing them to span the embeddings generated by handcrafted hard prompts maximally. To achieve this, we consider a set of handcrafted prompts (grammatically plausible) to produce class-specific embeddings $\{\boldsymbol{\omega}_c^m\}_{m=1}^{M}$, where $M$ is the number of hand-crafted textual prompts, and $c$ corresponds to class $c$. Then, the regularization is defined as follows

$$\mathcal{L}_{reg} = -\sum_{m=1}^{M} \sum_{c=1}^{C} \text{sim}(\boldsymbol{w}_c^m, \boldsymbol{U}_c), \tag{7}$$

where $\text{sim}(\cdot, \cdot)$ remains consistent with Eq. 4. Consequently, the total loss for optimizing our model is

$$\mathcal{L} = \mathcal{L}_{ce} + \gamma \mathcal{L}_{reg}, \tag{8}$$

where $\gamma$ controls the strength of the regularization.

**Ensembling of Linear Subspaces** Unlike some existing methods (Yao et al., 2023; Zhu et al., 2023), which only use a single or few hard prompts for regularizing the soft prompts, there is no restriction about how many hard prompts can be used in Eq. 7. Including more hard prompts in the regularization may facilitate maintaining the better zero-shot inference yet potentially introduce the difficulty of modelling an individual linear subspace to span all of them. To this end, one can enlarge the complexity of the modelled subspace, e.g. make it quadratic. However, that may make the point-to-subspace distance complex to compute and lose the chance of finding an analytic closed-form solution. Instead, we propose an ensembling of linear subspaces to cope with this situation when many hard prompts are proposed. Specifically, we model $N$ different subspaces and use $N$ groups of hard prompts with similar constraints per group to regularize them separately. Now the predictive probability of belonging to class $c$ for an image feature $\boldsymbol{f}$ is calculated as

$$p(c|\boldsymbol{f}) = \frac{1}{N} \sum_{j=1}^{N} \frac{\exp\left(\text{sim}(\boldsymbol{f}, \boldsymbol{U}_c^j)/\tau\right)}{\sum_{i=1}^{C} \exp\left(\text{sim}(\boldsymbol{f}, \boldsymbol{U}_i^j)/\tau\right)}. \tag{9}$$

---

[3]One can use the point-to-subspace distance for computing the predictive probability. We simply use this similarity measure retaining using a sim(,) function.

And the regularization in Eq.7 becomes

$$\mathcal{L}_{reg} = -\sum_{j=1}^{N} \sum_{m=1}^{\frac{M}{N}} \sum_{c=1}^{C} \text{sim}(\{\boldsymbol{w}_c^m\}_j, \boldsymbol{U}_c^j), \tag{10}$$

where $\boldsymbol{U}_c^j$ denotes the subspace derived by the set $j$ of soft prompts and $\{\boldsymbol{w}_c^m\}_j$ the set $j$ of hard prompts[4]. Basically, we train $N$ separate sets of soft prompts using Eq. 8 by substituting Eq. 6 with Eq. 9 and Eq. 7 with Eq. 10. Note that although this is an ensembling model, the pre-trained CLIP remains one copy, and only the size of parameter-efficient soft prompts gets scaled $N$ times up, which is still tiny compared to the existing priors. We name this variant SuPr-Ens.

## 4 EXPERIMENTS

We present experimental evaluations in this section. Our setup follows the protocols of CoCoOp (Zhou et al., 2022a) and LASP (Bulat & Tzimiropoulos, 2023), covering tasks such as generalization from base-to-new classes, domain generalization,cross-dataset transfer and few-shot learning, ensuring a thorough evaluation of our approach. Overall, the experimental results consistently showcase the outstanding performance of our method across all experimental settings.

**Datasets** Our experiments employed 11 diverse image classification datasets, encompassing a wide spectrum of visual recognition tasks. These datasets include ImageNet (Deng et al., 2009) and Caltech101 (Fei-Fei, 2004) for generic object classification, OxfordPets (Parkhi et al., 2012), StanfordCars (Krause et al., 2013), Flowers102 (Nilsback & Zisserman, 2008), Food101 (Bossard et al., 2014), and FGVCAircraft (Maji et al., 2013) for fine-grained visual recognition, EuroSAT (Helber et al., 2019) for satellite image classification, UCF101 (Soomro et al., 2012) for action recognition, DTD (Cimpoi et al., 2014) for texture classification, and SUN397 (Xiao et al., 2010) for scene recognition. For domain generalization experiments, we treated ImageNet as the source domain and evaluated methods on its variants as target domains, including ImageNetV2 (Recht et al., 2019), ImageNet-Sketch (Wang et al., 2019), ImageNet-A (Hendrycks et al., 2021b), and ImageNet-R (Hendrycks et al., 2021a).

**Implementation** Our implementation builds on the CoCoOp and LASP codebases, with CLIP serving as the foundation model. We follow the same training procedures, including using the Vit-B/16 vision backbone, data augmentation, and a fixed context length of $L = 4$. We report the final performance averaged over three random seeds. In Eq. 7, the number of textual templates $M$ was set to 60. Among these templates, 30 are drawn from CLIP (consistent with LASP), serving as dataset-agnostic templates, while the remaining 30 are generated from Chat-GPT, tailored to the specific dataset. For our ensembling model, the number of subspaces $N$ is set to three, and hyperparameter $\gamma$ in Eq. 8 is consistently set to 5 across all datasets.

**Baselines** We compare our method with several prior arts. The pre-trained CLIP (Radford et al., 2021) (ICML21). CoOp (Zhou et al., 2022b) (IJCV22) which uses soft prompts. CoCoOp (Zhou et al., 2022a) (CVPR22), which improves CoOp by adding image-conditional prompts. ProDA (Lu et al., 2022) (CVPR22), which considers soft prompt as Gaussian Distribution. ProGrad (Zhu et al., 2023) (ICCV22), which optimizes prompts to align with the "general direction". TaskRes (Yu et al., 2023) (CVPR23), which optimizes task-specific residual in embedding space, and KgCoOp (Yao et al., 2023) (CVPR23), which optimizes prompts close to fixed prompts in CLIP. LASP (Bulat & Tzimiropoulos, 2023) (CVPR23), which focuses on language-only optimization for downstream adaptation. And our SuPr and its ensembling variant SuPr-Ens.

### 4.1 GENERALIZATION FROM BASE-TO-NEW CLASSES

In this setting, each of the eleven datasets used has two splits: base and new classes, without overlap. Methods are trained on base classes and tested on the test sets of both base and novel classes as per CoCoOp.

Results in Table 1 show that our SuPr and SuPr-Ens achieve the best results across the majority of datasets. Compared with ProGrad and KgCoOp, which rely on strong alignment between soft

---

[4]Displayed in detail in the supplementary

Table 1: Comparison with the state-of-the-art on 11 datasets. Δ denotes the absolute improvement of our best variant, SuPr-Ens, over the previous best result LASP. Values in the brackets show the delta performance between each baseline and SuPr-Ens.

| Dataset | Set | CLIP | CoOp | CoCoOp (CVPR22) | ProGrad (ICCV23) | ProDA (CVPR22) | KgCoOp (CVPR23) | LASP (CVPR23) | SuPr | SuPr Ens | Δ |
|---|---|---|---|---|---|---|---|---|---|---|---|
| Average | Base | 69.34 | 82.69 | 80.47 (2.07 ↑) | 82.48 (0.06 ↑) | 81.56 (0.98 ↑) | 80.73 (1.81 ↑) | 83.10 (0.56 ↓) | 81.47 | 82.54 | −0.56 |
| | New | 74.22 | 63.22 | 71.69 (4.67 ↑) | 70.74 (5.62 ↑) | 72.30 (3.98 ↑) | 73.61 (2.75 ↑) | 76.11 (0.25 ↑) | 75.21 | 76.36 | +0.25 |
| | H | 71.70 | 71.66 | 75.83 (3.50 ↑) | 76.16 (3.17 ↑) | 76.65 (2.68 ↑) | 77.01 (2.32 ↑) | 79.45 (0.12 ↓) | 78.21 | 79.33 | −0.12 |
| ImageNet | Base | 72.43 | 76.47 | 75.98 (0.72 ↑) | 77.02 (0.32 ↑) | 75.40 (1.30 ↑) | 75.83 (0.87 ↑) | 76.25 (0.45 ↑) | 76.70 | 76.70 | +0.45 |
| | New | 68.14 | 67.88 | 70.43 (0.57 ↑) | 66.66 (4.34 ↑) | 70.23 (0.77 ↑) | 69.96 (1.04 ↑) | 71.17 (0.17 ↓) | 69.60 | 71.00 | −0.17 |
| | H | 70.22 | 71.92 | 73.10 (0.64 ↑) | 71.47 (2.27 ↑) | 72.72 (1.02 ↑) | 72.78 (0.96 ↑) | 73.62 (0.12 ↑) | 72.98 | 73.74 | +0.12 |
| Caltech101 | Base | 96.84 | 98.00 | 97.96 (0.07 ↑) | 98.02 (0.01 ↑) | 98.27 (0.24 ↓) | 97.72 (0.31 ↑) | 98.17 (0.14 ↓) | 98.00 | 98.03 | −0.14 |
| | New | 94.00 | 89.91 | 93.81 (1.02 ↑) | 93.89 (0.94 ↑) | 93.23 (1.60 ↑) | 94.39 (0.44 ↑) | 94.33 (0.50 ↑) | 94.30 | 94.83 | +0.50 |
| | H | 95.40 | 93.73 | 95.84 (0.56 ↑) | 95.91 (0.49 ↑) | 95.86 (0.54 ↑) | 96.03 (0.37 ↑) | 96.21 (0.19 ↑) | 96.11 | 96.40 | +0.19 |
| OxfordPets | Base | 91.17 | 93.67 | 95.20 (0.70 ↑) | 95.07 (0.83 ↑) | 95.43 (0.47 ↑) | 94.65 (1.25 ↑) | 95.73 (0.17 ↑) | 95.37 | 95.90 | +0.17 |
| | New | 97.26 | 95.29 | 97.69 (0.29 ↓) | 97.63 (0.23 ↓) | 97.83 (0.43 ↓) | 97.76 (0.36 ↓) | 97.87 (0.47 ↓) | 97.00 | 97.40 | −0.47 |
| | H | 94.12 | 94.47 | 96.43 (0.21 ↑) | 96.33 (0.31 ↑) | 96.62 (0.02 ↑) | 96.18 (0.46 ↑) | 96.79 (0.15 ↓) | 96.18 | 96.64 | −0.15 |
| StanfordCars | Base | 63.37 | 78.12 | 70.49 (4.94 ↑) | 77.68 (2.25 ↓) | 74.70 (0.73 ↑) | 71.76 (3.67 ↑) | 75.23 (0.20 ↑) | 72.90 | 75.43 | +0.20 |
| | New | 74.89 | 60.40 | 73.59 (1.14 ↑) | 68.63 (6.10 ↑) | 71.20 (3.53 ↑) | 75.04 (0.31 ↓) | 71.77 (2.96 ↑) | 75.13 | 74.73 | +2.96 |
| | H | 68.85 | 68.13 | 72.01 (3.07 ↑) | 72.88 (2.20 ↑) | 72.91 (2.17 ↑) | 73.36 (1.72 ↑) | 73.46 (1.62 ↑) | 74.00 | 75.08 | +1.62 |
| Flowers102 | Base | 72.08 | 97.60 | 94.87 (2.63 ↑) | 95.54 (1.96 ↑) | 97.70 (0.20 ↓) | 95.00 (2.42 ↑) | 97.17 (0.33 ↑) | 96.17 | 97.50 | +0.33 |
| | New | 77.80 | 59.67 | 71.75 (4.45 ↑) | 71.87 (4.33 ↑) | 68.68 (7.52 ↑) | 74.73 (1.47 ↑) | 73.53 (2.75 ↑) | 75.93 | 76.20 | +2.67 |
| | H | 74.83 | 74.06 | 81.71 (3.83 ↑) | 82.03 (3.51 ↑) | 80.66 (4.88 ↑) | 83.65 (1.89 ↑) | 83.71 (1.83 ↑) | 85.19 | 85.54 | +1.83 |
| Food101 | Base | 90.10 | 88.33 | 90.70 (0.23 ↑) | 90.37 (0.56 ↑) | 90.30 (0.63 ↑) | 90.50 (0.43 ↑) | 91.20 (0.27 ↓) | 90.77 | 90.93 | −0.27 |
| | New | 91.22 | 82.26 | 91.29 (0.88 ↑) | 89.59 (2.58 ↑) | 88.57 (3.60 ↑) | 91.70 (0.47 ↑) | 91.90 (0.27 ↑) | 92.13 | 92.17 | +0.27 |
| | H | 90.66 | 85.19 | 90.99 (0.56 ↑) | 89.98 (1.57 ↑) | 89.43 (2.12 ↑) | 91.10 (0.45 ↑) | 91.54 (0.01 ↑) | 91.44 | 91.55 | +0.01 |
| FGVCAircraft | Base | 27.19 | 40.44 | 33.41 (3.82 ↑) | 40.54 (3.31 ↓) | 36.90 (0.33 ↑) | 36.21 (1.02 ↑) | 38.05 (0.82 ↓) | 37.00 | 37.23 | −0.82 |
| | New | 36.29 | 22.30 | 23.71 (13.2 ↑) | 27.57 (9.30 ↑) | 34.13 (2.74 ↑) | 33.55 (3.32 ↑) | 33.20 (3.67 ↑) | 34.13 | 36.87 | +3.67 |
| | H | 31.09 | 28.75 | 27.74 (9.31 ↑) | 32.82 (4.23 ↑) | 35.46 (1.59 ↑) | 34.83 (2.22 ↑) | 35.46 (1.59 ↑) | 35.51 | 37.05 | +1.59 |
| SUN397 | Base | 69.36 | 80.60 | 79.74 (2.56 ↑) | 81.26 (1.04 ↑) | 78.67 (3.63 ↑) | 80.29 (2.01 ↑) | 80.70 (1.60 ↑) | 81.70 | 82.30 | +1.60 |
| | New | 75.35 | 65.89 | 76.86 (1.94 ↑) | 74.17 (4.63 ↑) | 76.93 (1.87 ↑) | 76.53 (2.27 ↑) | 79.30 (0.50 ↓) | 78.80 | 78.80 | −0.50 |
| | H | 72.23 | 72.51 | 78.27 (2.24 ↑) | 77.55 (2.96 ↑) | 77.79 (2.72 ↑) | 78.36 (2.15 ↑) | 80.00 (0.51 ↑) | 80.22 | 80.51 | +0.51 |
| DTD | Base | 53.24 | 79.44 | 77.01 (4.26 ↑) | 77.35 (3.92 ↑) | 80.67 (0.60 ↑) | 77.55 (3.72 ↑) | 81.10 (0.17 ↑) | 78.20 | 81.27 | +0.17 |
| | New | 59.90 | 41.18 | 56.00 (6.70 ↑) | 52.35 (10.4 ↑) | 56.48 (6.22 ↑) | 54.99 (7.71 ↑) | 62.57 (0.13 ↑) | 61.07 | 62.70 | +0.13 |
| | H | 56.37 | 54.24 | 64.85 (5.94 ↑) | 62.44 (8.35 ↑) | 66.44 (4.35 ↑) | 64.35 (6.44 ↑) | 70.64 (0.15 ↑) | 68.58 | 70.79 | +0.15 |
| EuroSAT | Base | 56.48 | 92.19 | 87.49 (0.44 ↑) | 90.11 (2.18 ↓) | 83.90 (4.03 ↑) | 85.64 (2.29 ↑) | 95.00 (7.07 ↓) | 86.07 | 87.93 | −7.07 |
| | New | 64.05 | 54.74 | 60.04 (16.1 ↑) | 60.89 (15.2 ↑) | 66.00 (10.1 ↑) | 64.34 (11.8 ↑) | 83.37 (7.27 ↓) | 71.83 | 76.10 | −7.27 |
| | H | 60.03 | 68.90 | 71.21 (10.4 ↑) | 72.67 (8.92 ↑) | 73.88 (7.71 ↑) | 73.48 (8.11 ↑) | 88.81 (7.22 ↓) | 78.31 | 81.59 | −7.22 |
| UCF101 | Base | 70.53 | 84.69 | 82.33 (2.44 ↑) | 84.33 (0.44 ↑) | 85.23 (0.46 ↓) | 82.89 (1.88 ↑) | 85.53 (0.76 ↓) | 83.33 | 84.77 | −0.76 |
| | New | 77.50 | 56.05 | 73.45 (5.68 ↑) | 74.94 (4.19 ↑) | 71.97 (7.16 ↑) | 76.67 (2.46 ↑) | 78.20 (0.93 ↑) | 76.47 | 79.13 | +0.93 |
| | H | 73.85 | 67.46 | 77.64 (4.21 ↑) | 79.36 (2.49 ↑) | 78.04 (3.81 ↑) | 79.66 (2.19 ↑) | 81.70 (0.15 ↑) | 79.75 | 81.85 | +0.15 |

Table 2: a) Domain Generalization and b) Cross-dataset transfer.

(a) Comparison of prompt learning in domain generalization with 16-shot source samples. The best results are marked **bold**.

(b) Comparison in the cross-dataset transfer learning by learning the prompts from ImageNet (16-shots) with ViT/B16, and evaluating on the other 10 datasets. The best results are marked **bold**. * reimplemented.

| | Source | Target | | | | Avg. |
|---|---|---|---|---|---|---|
| | ImageNet | ImageNetV2 | ImageNet-Sketch | ImageNet-A | ImageNet-R | |
| CLIP | 66.73 | 60.83 | 46.15 | 47.77 | 73.96 | 57.17 |
| CoCoOp | 71.02 | 64.07 | 48.75 | 50.63 | 76.18 | 59.90 |
| CoOp | 71.51 | 64.20 | 47.99 | 49.71 | 75.21 | 59.28 |
| ProGrad | 72.24 | 64.73 | 47.61 | 49.39 | 74.58 | 59.07 |
| KgCoOp | 71.20 | 64.10 | 48.97 | 50.69 | 76.70 | 60.11 |
| TaskRes | 73.90 | **65.85** | 47.70 | 49.17 | 75.23 | 59.49 |
| LASP | 71.10 | 63.96 | 49.01 | **50.70** | 77.07 | 60.19 |
| SuPr | 71.70 | 64.50 | **49.40** | 50.30 | **77.20** | **60.35** |

| | Datasets | CoOp | CoCoOp | ProGrad | KgCoOp | LASP* | SuPr |
|---|---|---|---|---|---|---|---|
| Source | ImageNet | 71.51 | 71.02 | 72.24 | 70.66 | 71.40 | 71.70 |
| Target | Caltech101 | 93.70 | **94.43** | 91.52 | 93.92 | 93.30 | 94.20 |
| | OxfordPets | 89.14 | **90.14** | 89.64 | 89.83 | 89.88 | 89.80 |
| | StanfordCars | 64.51 | 65.32 | 62.39 | **65.41** | 65.01 | 64.90 |
| | Flowers102 | 68.71 | 71.88 | 67.87 | 70.01 | 70.20 | **70.70** |
| | Food101 | 85.30 | 86.06 | 85.40 | **86.36** | 85.39 | 86.30 |
| | FGVCAircraft | 18.47 | 22.94 | 20.61 | 22.51 | 20.88 | **23.00** |
| | SUN397 | 64.15 | **67.36** | 62.47 | 66.16 | 66.74 | 66.50 |
| | DTD | 41.92 | 45.73 | 39.42 | **46.35** | 43.67 | 45.50 |
| | EuroSAT | 46.39 | 45.37 | 43.46 | 46.04 | 45.32 | **50.20** |
| | UCF101 | 66.55 | 68.21 | 64.29 | 68.50 | **69.07** | 67.70 |
| | Avg. | 63.88 | 65.74 | 62.71 | 65.51 | 64.95 | **65.88** |

and hard prompts, our SuPr relaxes this strong regularization and outperforms them in both the base- and novel-class results. LASP is a strong competitor, considering the overall average performance. However, its primary improvement comes from the EuroSat dataset, without which our approach outperforms LASP significantly (see Figure 3a). Notably, our method excels on fine-grained datasets, emphasizing the effectiveness of subspace modelling in capturing diverse underlying semantic concepts, which greatly benefit fine-grained recognition tasks.

## 4.2 DOMAIN GENERALIZATION & CROSS DATASET TRANSFER

To further assess the robustness of our method, we perform experiments in domain generalization and cross-dataset transfer scenarios. In both settings, methods are trained on ImageNet as the source domain. However, they are evaluated on ImageNet variants in Table 2a in a closed-set setting and on novel datasets in Table 2b in an open-set setting.

It is seen in Tables 2a&2b, our approach consistently demonstrates excellent performance on target domains in both settings, emphasizing its impressive generalization capabilities. In domain generalization, we achieve the best results in two out of four domains, with the highest overall average performance. In cross-dataset transfer, we again outperform other methods by having the best average

Table 3: a) Few-shot learning within base classes. b) Ablation study.

(a) Accuracy (%) of few-shot (4 shots) learning on 11 datasets using the ViT/B16 backbone. The best results are marked **bold**. * reimplemented.

(b) Ablation study for each component.

| Datasets | CoOp | CoCoOp | ProGrad | KgCoOp | LASP* | SuPr |
|---|---|---|---|---|---|---|
| ImageNet | 69.38 | **70.55** | 70.21 | 70.19 | 70.47 | 69.77 |
| Caltech101 | 94.44 | 94.98 | 94.93 | 94.65 | 94.70 | **95.17** |
| OxfordPets | 91.30 | 93.01 | **93.21** | 93.20 | 92.58 | 93.13 |
| StanfordCars | 72.73 | 69.10 | 71.75 | 71.98 | 71.97 | **76.80** |
| Flowers102 | 91.14 | 82.56 | 89.98 | 90.69 | 89.48 | **94.23** |
| Food101 | 82.58 | **86.64** | 85.77 | 86.59 | 85.85 | 86.00 |
| FGVCAircraft | 33.18 | 30.87 | 32.93 | 32.47 | 30.60 | **35.53** |
| SUN397 | 70.13 | 70.50 | 71.17 | 71.79 | 72.32 | **73.60** |
| DTD | 58.57 | 54.79 | 57.72 | 58.31 | 58.39 | **64.97** |
| EuroSAT | 68.62 | 63.83 | 70.84 | 71.06 | 68.80 | **73.23** |
| UCF101 | 77.41 | 74.99 | 77.82 | 78.40 | 78.24 | **79.97** |
| Avg. | 73.59 | 71.98 | 74.21 | 74.48 | 73.95 | **76.58** |

| Set | CoOp | CoOp Ensemble | SuPr w/o reg | SuPr | SuPr Ens |
|---|---|---|---|---|---|
| Multiple Prompts | ✗ | ✓ | ✓ | ✓ | ✓ |
| Subspace Modeling | ✗ | ✗ | ✓ | ✓ | ✓ |
| Regularization | ✗ | ✗ | ✗ | ✓ | ✓ |
| Subspace Ensemble | ✗ | ✗ | ✗ | ✗ | ✓ |
| Base | 82.69 | 80.02 | 81.18 | 81.47 | 82.54 |
| New | 63.22 | 68.51 | 73.30 | 75.21 | 76.36 |
| H | 71.66 | 73.82 | 77.04 | 78.21 | 79.33 |

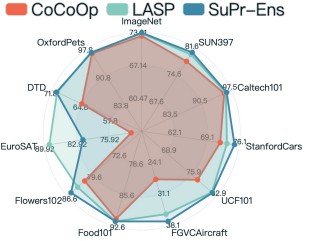

(a) Harmonic mean.

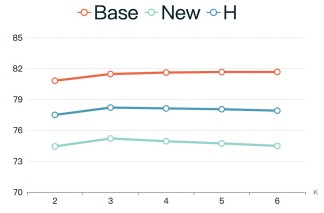

(b) Soft prompts number $K$.

Figure 3: a) overall performance of Table 1 and b) varying $K$.

performance. These findings highlight the robustness of our method and its superior generalization capabilities across various datasets compared to point-estimate methods.

## 4.3 FEW SHOT LEARNING WITHIN BASE CLASSES

In addition, we conduct experiments in the few-shot learning setting (only base class) following (Yao et al., 2023). Unlike other methods that compromise their performance on base classes due to strong regularization, our method demonstrates a remarkable average performance improvement of 2.1% across the eleven datasets. Notably, our approach outperforms existing methods on eight of the eleven datasets in this particular setting, as detailed in Table 3a. Again, these results underscore the versatility of our approach, which maintains superior performance on both base and new classes.

## 4.4 ABLATION STUDY AND FURTHER ANALYSIS

We conduct an ablation study to analyse each component in our proposed SuPr and SuPr-Ens in Table 3b[5]. We also randomly select six classes from the Food101 dataset and perform t-SNE visualizations. Figure 4a/b illustrates the t-SNE visualizations for novel classes, while Figure 4c for base classes. Image features, soft- and hard-prompt embeddings are represented by circles, inverted triangles, and pentagrams. Different colours indicate distinct classes. Furthermore, we employ *least square* to fit linear subspaces to the soft-prompt embeddings, resulting in linear planes corresponding to each class, visually represented by large squares.

**Ablation study.** From the results in Table 3b, it is seen that CoOp-Ensemble improves CoOp with about 2.16% accuracy by learning separately multiple learnable sets of soft prompts and ensembling the learned CoOp models. Incorporating the subspace modelling on top of CoOp-Ensemble to replace the simple ensembling improves its performance further by 3.22%, evidencing the ef-

---

[5]**CoOp:** Vanilla CoOp, assuming the parameter size of learnable soft prompts is $M$. **CoOp-Ensemble:** Multiple CoOp models are introduced, i.e., multiple sets of soft prompts are learnable, whose total parameter size is also $M$. During inference, prediction is based on the ensembling of multiple CoOp models. **SuPr w/o reg:** Adding subspace modeling on top of CoOp-Ensemble. **SuPr:** Adding hard-prompt based regularization on top of SuPr w/o reg. **SuPr-Ens:** Ensembling separate linear subspaces, which are regularized by different subsets of hard prompts for a class.

Figure 4: t-SNE visualization.

fectiveness of SuPr w/o reg. Furthermore, adding the hard-prompt-based regularization pushes the model performance up by $1.17\%$, as the achievement of SuPr, which is further improved with $1.12\%$ accuracy by subspace ensembling – SuPr-Ens.

**What is the effect of varying $K$ of soft prompts?** In Figure 3b, we conduct experiments to evaluate the impact of varying the number of sets of soft prompts. Our observations reveal that the overall performance remains relatively stable as $K$ changes. Using more soft prompts slightly improves base class accuracy but decreases accuracy for novel classes. The optimal performance is achieved when $K$ is set to three, which is used in all experiments.

**How does hard-prompt regularization help?** From the comparison between SuPr w/o reg and SuPr in Table 3b, we can see a notable improvement brought by the regularization. The reason is reflected in Figure 4a. We can see in Figure 4a that SuPr w/o reg does not generalize to novel classes since the image features are not associated with the correct subspaces. However, SuPr w/reg mitigates this issue noticeably and now the image features can be assigned to the right subspaces. Figure 4b explains it. We can see that after regularization, the modelled subspaces get tuned to align better hard-prompt embeddings, leading to better zero-shot inference on novel classes.

**What is the benefit of SuPr-Ens compared with SuPr?** We conduct t-SNE visualizations for both SuPr and SuPr-Ens (two subspaces per class). Through Figure 4c, we observe that when only five hard prompts are used the modelled subspace can span them well in Figure 4c (left), but when the number of hard prompts goes up to thirty, spanning all of them on a single subspace is hard as shown in Figure 4c (mid). However, using the same hard prompts, the accommodation of hard-prompt embeddings becomes less difficult when multiple subspaces are introduced, as shown in Figure 4c (right). Different subspaces can embed different subsets of the hard prompts, capturing different spectrums of textual/visual details.

## 5 CONCLUSION

We introduced SuPr, a linear subspace-based prompt learning method for adapting vision-language models in this paper. Our approach marks the first integration of subspace modelling with VLMs, allowing the model to encompass a wider spectrum of visual semantics. Our proposed subspace modelling addresses the challenge of base class overfitting by accommodating hard prompts within the subspace modelled by the learned soft prompts. SuPr excels in fitting to the base classes while generalizing to novel classes. Our SuPr achieved strong performance on various tasks, including generalization from base to novel, domain generalization, cross-dataset transfer and few shot learning. Interestingly, a simple ensembling of modelling multiple linear subspaces using different sets of hard prompts results in a more leading variant, achieving state of the art performance on all settings.

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

# A  HARD PROMPTS

## Dataset Specific Prompt Template

ImageNet
"`a photo of a {CLASS}.`"
"`itap of a {CLASS}.`"
"`a bad photo of the {CLASS}.`"
"`a origami {CLASS}.`"
"`a photo of the large {CLASS}.`"
`...`

Caltech101
"`a photo of a {CLASS}.`"
"`a painting of a {CLASS}.`"
"`a plastic {CLASS}.`"
"`a sculpture of a {CLASS}.`"
"`a sketch of a {CLASS}.`"
`...`

OxfordPets
"`a photo of a {CLASS}, a type of pet.`"
"`a photo of a {CLASS}, a type of cat.`"
"`a photo of a {CLASS}, a type of dog.`"
"`a photo of a {CLASS} pet.`"
"`a picture of a {CLASS} animal companion.`"
`...`

StanfordCars
"`a photo of a {CLASS}.`"
"`a photo of the {CLASS}.`"
"`a photo of my {CLASS}.`"
"`I love my {CLASS}.`"
"`a photo of my dirty {CLASS}.`"
`...`

Flowers102
"`a photo of a {CLASS} flower.`"
"`a photo of a {CLASS} bloom.`"
"`a picture of a {CLASS} flower.`"
"`an image of a {CLASS} bloom .`"
"`a close-up of a {CLASS} flower.`"
`...`

Food101
"`a photo of {CLASS}, a type of food.`"
"`a photo of a {CLASS}, a type of food.`"
"`a photo of the {CLASS}, a type of food.`"
"`a photo of a plate of {CLASS}.`"
"`a picture of a dish of {CLASS}.`"
`...`

| | |
|---|---|
| FGVCAircraft | "a photo of a {CLASS}, a type of aircraft." |
| | "a photo of the {CLASS}, a type of aircraft." |
| | "a photo of a {CLASS} airliner." |
| | "a picture of a {CLASS} passenger plane." |
| | "an image of a {CLASS} commercial aircraft." |
| | ... |
| SUN397 | "a photo of a {CLASS}." |
| | "a photo of the {CLASS}." |
| | "itap of a {CLASS}." |
| | "a bad photo of the {CLASS}." |
| | "a origami {CLASS}." |
| | ... |
| DTD | "{CLASS} texture." |
| | "a photo of a {CLASS} texture." |
| | "a photo of a {CLASS} pattern." |
| | "a photo of a {CLASS} thing." |
| | "a photo of a {CLASS} object." |
| | ... |
| EuroSAT | "a centered satellite photo of {CLASS}." |
| | "a centered satellite photo of a {CLASS}." |
| | "a centered satellite photo of the {CLASS}." |
| | "an aerial view showcasing {CLASS}." |
| | "a satellite image capturing {CLASS}." |
| | ... |
| UCF101 | "a photo of a person doing {CLASS}." |
| | "a photo of a person {CLASS}." |
| | "a video of a person {CLASS}." |
| | "a example of a person {CLASS}." |
| | "a demonstration of a person {CLASS}." |
| | ... |

## B  QUALITATIVE VISUALIZATION

**Analysis on subspace.**    We conduct the qualitative visualization using Paella Rampas et al. (2023) to synthesize images based on different (hard & learned) prompts. The visualizations in Figure 5-8 show that soft prompts learned by subspace modelling capture different intra-class variations, such as fine-grained attributes in terms of colour, texture and depiction styles. This explains why our SuPr improves over CoOp, which is stuck with learning only dominating concepts. Also, walking in the subspace across different subspace bases shows interesting transitions along different attributes, showing the wealth of semantic information learned in each subspace, as shown in Figure 9-11.

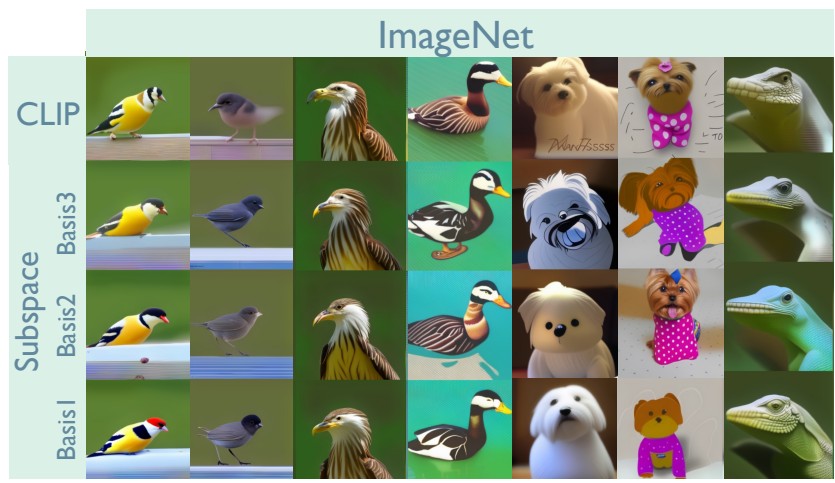

Figure 5: Text to image synthesis using different prompts on ImageNet.

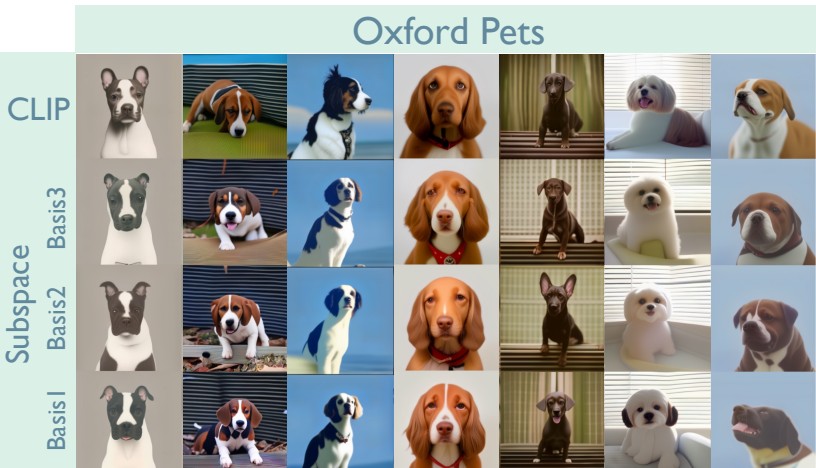

Figure 6: Text to image synthesis using different prompts on Oxford Pet.

**How do subspaces enable better inference compared with vector classifiers?** We compute the prediction scores for all the test samples for some datasets and visualize the top 10% prediction-confident samples in Figure 12-14. To better understand the selected samples, we cluster them into three clusters using K-means. From the results, we can see that the test samples that simulate the vector classifiers are less diverse than the subspace ones, indicating the issue of learning only the dominating concepts of vector classifiers. Among them, we can also see the samples predicted right using subspace classifiers but not by vector classifiers.

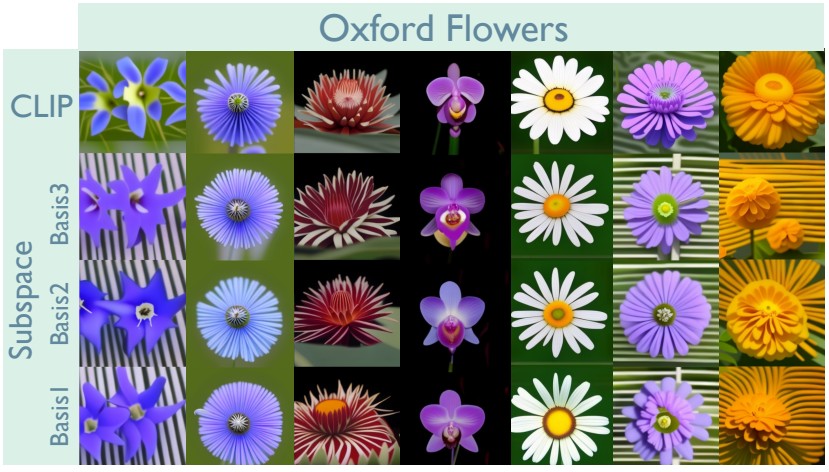

Figure 7: Text to image synthesis using different prompts on Oxford Flower.

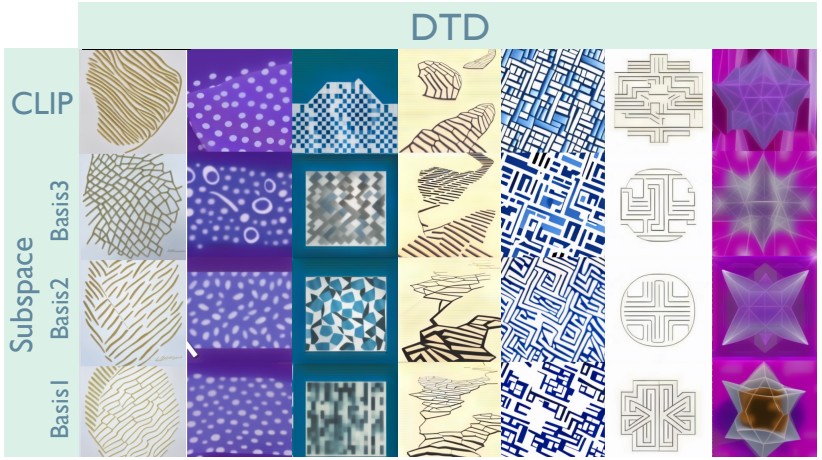

Figure 8: Text to image synthesis using different prompts on DTD.

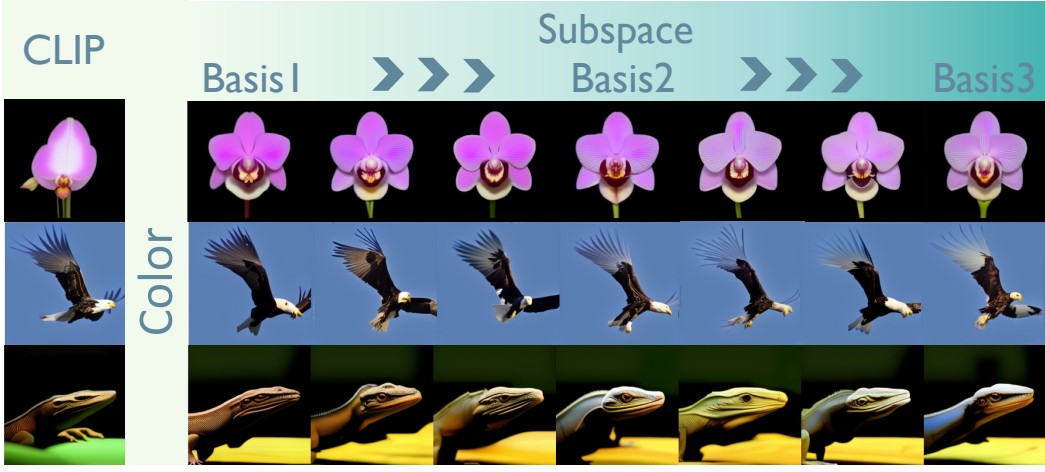

Figure 9: Subspace walking — color attribute.

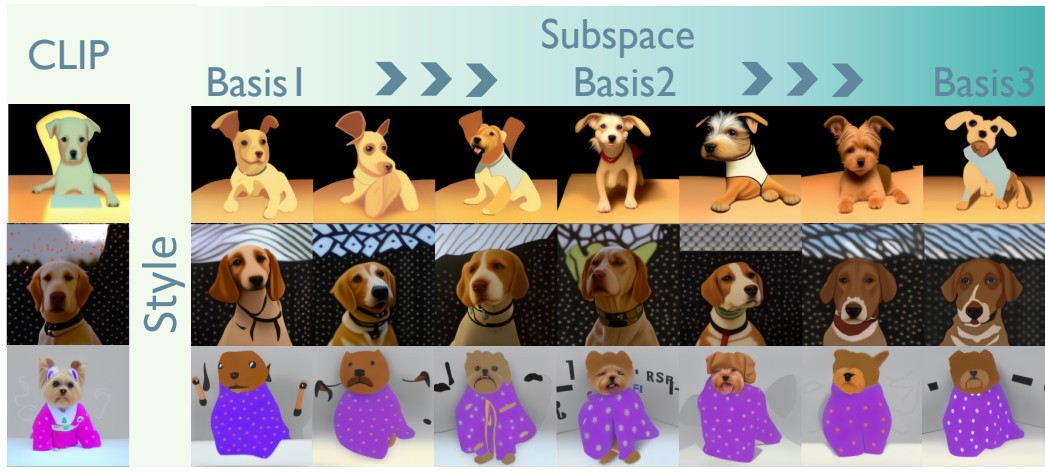

Figure 10: Subspace walking — texture attribute.

Figure 11: Subspace walking — depiction styles.

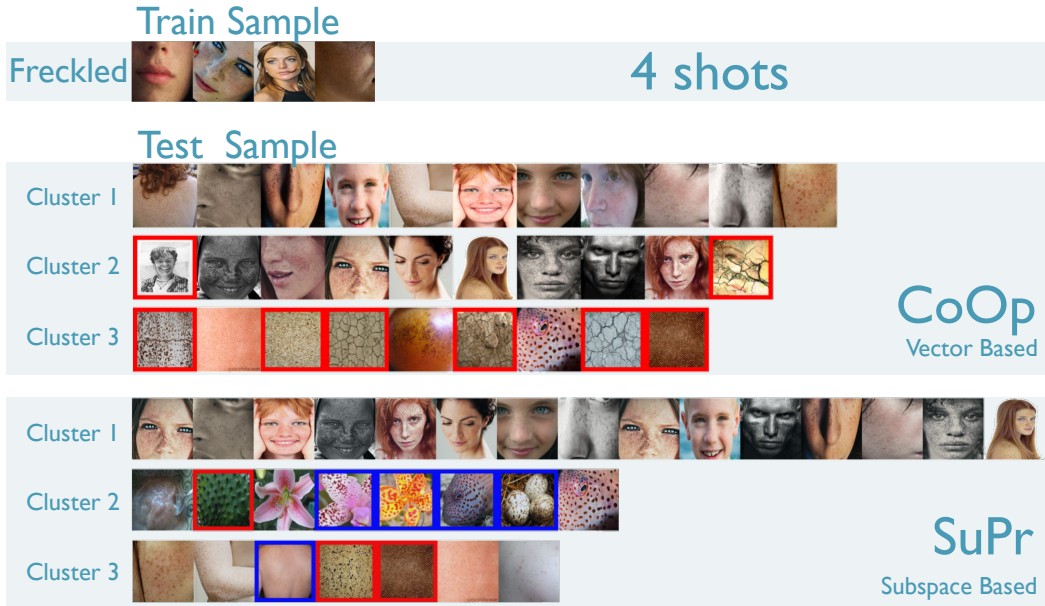

Figure 12: Prediction-confident test samples of Freckled for vector v.s. subspace classifiers. Samples with red boxing are wrong predictions, and samples with blue boxing are predicted right with subspace classifiers but not with vector classifiers.

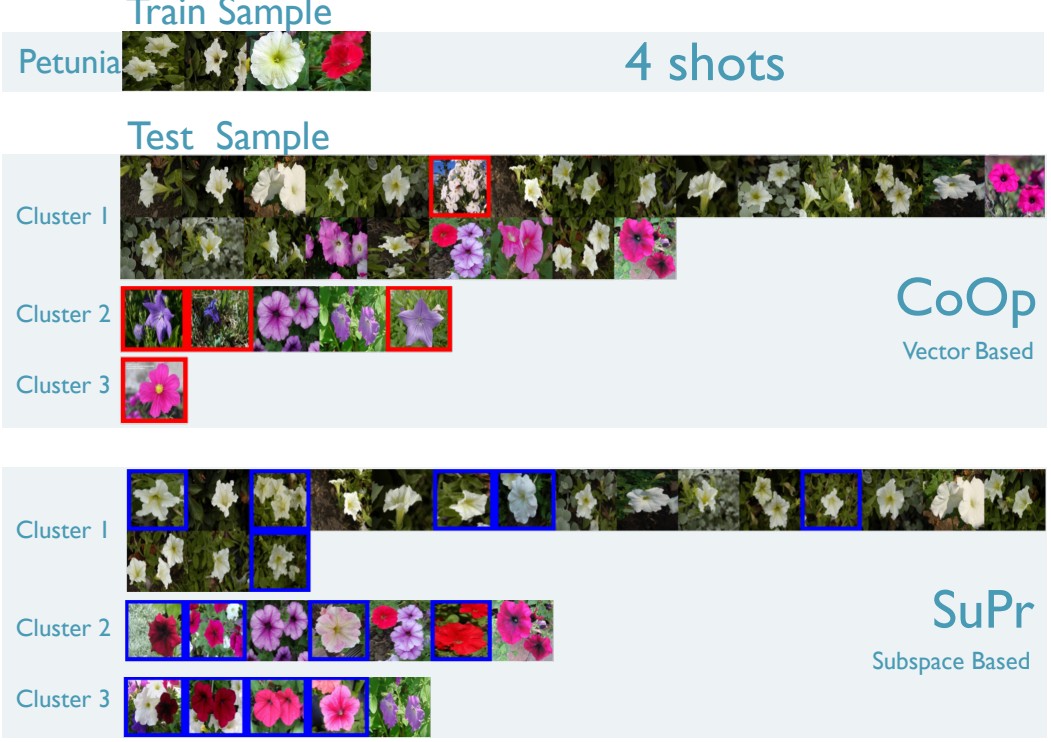

Figure 13: Prediction-confident test samples of Petunia for vector v.s. subspace classifiers. Samples with red boxing are wrong predictions, and samples with blue boxing are predicted right with subspace classifiers but not with vector classifiers.

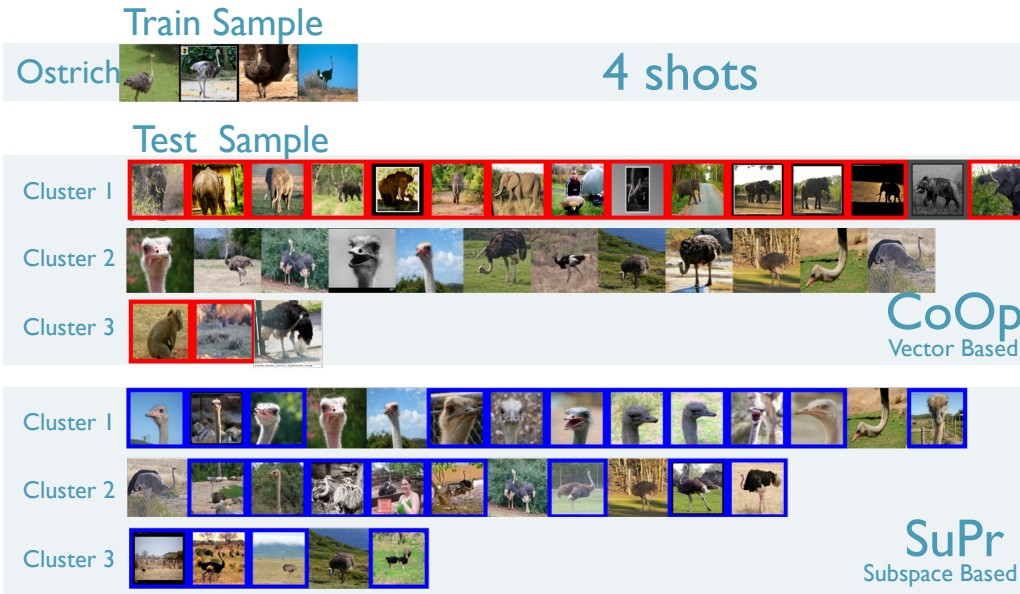

Figure 14: Prediction-confident test samples of Ostrich for vector v.s. subspace classifiers. Samples with red boxing are wrong predictions, and samples with blue boxing are predicted right with subspace classifiers but not with vector classifiers.

