# OpenReview forum: "Vision-Language Subspace Prompting"
_ICLR.cc/2024/Conference — Submitted to ICLR 2024_

### Official Review · Reviewer_nXYV · 2023-10-24

**Soundness:** 3 good
**Presentation:** 3 good
**Contribution:** 2 fair
**Rating:** 5
**Confidence:** 5

**Summary:**

This work focused on how to conduct prompt tuning on vision-language models (i.e., CLIP), and proposed a subspace-based prompt learning method that divided soft prompts with orthonormal subgroups, regularized by hard prompts. Experiments on base-to-new classes, domain generalization, and cross-dataset transfer settings show the effectiveness of the method.

**Strengths:**

+ The proposed method achieved competitive performance on base-to-new classes, domain generalization, and cross-dataset transfer settings.

+ The method is simple but effective, although some insights behind the method are not clear now.

**Weaknesses:**

- Analysis about "Is subspace modeling useful" in Section 4.4. The conclusion is obtained based on the comparisons between SuPr w/o reg, CoOp, and CoOp-Ensemble. It is not clear what are the detailed differences among the three methods, which is essential to understand whether the comparisons can lead to the conclusions, as the performance gain may come from other components.

- SVD for subspace modeling. It is a bit hard for me to understand the role of SVD in subspace modeling. According to Sec. 3.2, it seems that SVD is to guarantee that the matrix $U_c$ is an orthonormal matrix. If so, is it possible to only restrict $U_c$ to be orthonormal without the SVD operation? Also, it is interesting to know the ablation where $U_c$ is no longer an orthonormal matrix. In this potential ablation study, can we say the subspace are no longer disentangled/independent?

- Main technical contribution. It seems that the main messages of this work are (1) dividing soft prompts into subgroups, (2) regularizing soft prompts with hard prompts. There lack insights why the subgroup manner works beyond the technical tricks.

- Analysis on subspace. Does the subspace have any semantic information, or what does each subspace represent? That would contribute to explainability.

**Questions:**

Please see weaknesses for detailed comments.

---

> ### Author Response · Authors · 2023-11-21
>
> # Response to nXYV
>
> **W1: Analysis about "Is subspace modeling useful" in Section 4.4.** We understand this is a confusion about our ablation study. Please refer to the response to all for this concern.
>
> **W2: SVD for subspace modelling.** **1) Restrict $U_c$ orthonormal without SVD:**  Good point. SVD was used to ensure the orthonormality of $U_c$ during optimization. Placing this restriction on $U_c$ directly was also considered in our preliminary implementation. However, $U_c$ was formed by the soft-prompt embeddings, i.e., it is generated dynamically (as soft prompts update iteratively), making this constraint hard to be in place for $U_c$. We will keep investigating this in our future research.  **2) What happens if $U_c$ is not an orthonormal matrix?**  Indeed, having an orthonormal matrix $U_c$ for linear subspace modelling is unnecessary. We can use the vanilla support points from the soft-prompt embeddings and employ least square linear regression to model a linear subspace as per [1]. We tried this during development, and it performed similarly to SVD - just $0.23\$% weaker. Thus we stayed with SVD for simplicity and slightly better empirical performance. **3) The subspace is no longer disentangled/independent when $U_c$ is unconstrained?** Yes. However, we would like to clarify that the orthogonality constraint imposed by SVD affects the bases that define each linear subspace, but it does not affect whether the subspaces are orthogonal to each other. IE: There is currently no inter-subspace independence/orthogonality constraint. We also tried regularising the subspaces to be orthogonal to each other during development, but this negatively affected performance.
>
> [1] Jerome Friedman, Trevor Hastie, and Robert Tibshirani. The elements of statistical learning, 2001.
>
> |        Hos(%)          | ImageNet | Caltech101 | OxfordPets | StanfordCars | Flowers102 | Food101 | FGVCAircraft | SUN397 | DTD   | EuroSAT | UCF101 | Average |
> | ---------------- | -------- | ---------- | ---------- | ------------ | ---------- | ------- | ------------ | ------ | ----- | ------- | ------ | ------- |
> | SuPr-Ens w/o SVD | 73.33    | 96.41      | 96.28      | 75.10        | 85.47      | 91.51   | 37.28        | 80.38  | 70.54 | 81.59   | 80.48  | 79.11   |
> | SuPr-Ens         | 73.74    | 96.40      | 96.64      | 75.08        | 85.54      | 91.55   | 37.05        | 80.51  | 70.79 | 81.59   | 81.85  | 79.33   |

---

> ### Author Response · Authors · 2023-11-21
>
> **W3: Main technical contribution.** Please first refer to the response to Reviewer $kdeh$ regarding 'novelty in subspace modelling'. May the take-home messages be refined from our clarification.
>
> **Clarification for Message (1):**  'dividing soft prompts into subgroups' should not be treated as a main contribution, as this is also just what CoOp-Ensemble requires for modelling.  Our first main contribution is ***subspace modelling*** with multiple groups of soft prompts. No prior prompt-learning approaches considered subspace modeling to represent categories. This is a great contribution, as learning a subspace classifier leads to better extrapolation/generalization than a prior vector/prototype based class representations such as CoOp. We will show in our revision that a subspace classifier captures the intra-class variability for a class rather than a single dominating point, such that it improves generalization.
>
> **Clarification for Message (2):** 'regularizing soft prompts with hard prompts'  can be enabled in various ways, such as strong alignment between soft/hard prompts. Excessive alignment can harm an adapted VLM's performance on the base classes, as observed in [2,3]. However, our contribution is the specific approach to regularize the modelled soft-prompts-based linear subspaces by forcing them to span hard-prompt embeddings. Our improved VLMs can be tailored well for base classes while maintaining generalization on unseen classes.
>
> **Explanation for insights into why the proposed method works:** Please refer to the response to the following concern. Also, we compute the prediction scores for all the test samples for some datasets and visualize the top $10%$% prediction-confident samples in Figure12-14. To better understand the selected samples, we cluster them into three clusters using K-means. From the results, we can see that the test samples that simulate the vector classifiers are less diverse than the subspace ones, indicating the issue of learning only the dominating concepts of vector classifiers. Among them, we can also see the samples predicted right using subspace classifiers but not by vector classifiers.
> - Prediction-confident test samples of [Freckled](https://p.ipic.vip/uukjqh.png)/[Petunia](https://p.ipic.vip/bff5ba.png)/[Ostrich](https://p.ipic.vip/w6r9ed.png) for vector v.s. subspace classifiers. Samples with red boxing are wrong predictions, and samples with blue boxing are predicted right with subspace classifiers but not with vector classifiers.
>
> [2] Hantao Yao, Rui Zhang, and Changsheng Xu. Visual-language prompt tuning with knowledgeguided context optimization. In CVPR, 2023.
>
> [3] Beier Zhu, Yulei Niu, Yucheng Han, Yue Wu, and Hanwang Zhang. Prompt-aligned gradient for prompt tuning. In ICCV, 2023.

---

> ### Author Response · Authors · 2023-11-21
>
> **W4: Analysis on subspace.** Thanks for the suggestion! We have now included the qualitative visualization in the revision using Paella [4] to synthesize images based on different (hard \& learned) prompts. Please refer to Figure 5-11 in the appendix of the revision or the links below. The visualizations in Figure 5-8 show that soft prompts learned by subspace modelling capture different intra-class variations, such as fine-grained attributes in terms of colour, texture and depiction styles. This explains why our SuPr improves over CoOp, which is stuck with learning only dominating concepts. Also, walking in the subspace across different subspace bases shows interesting transitions along different attributes, showing the wealth of semantic information learned in each subspace, as shown in Figure 9-11.
>
> - [Text to image synthesis using different prompts.](https://p.ipic.vip/uermie.png)
>
> - [Subspace walking --- color attribute.](https://p.ipic.vip/o14ydn.png)
>
> - [Subspace walking --- texture attribute.](https://p.ipic.vip/udzhp4.png)
>
> - [Subspace walking --- depiction styles.](https://p.ipic.vip/xiu4kb.png)
>
>
> [4] Dominic Rampas, Pablo Pernias, Marc Aubreville. A novel sampling scheme for textand image-conditional image synthesis in quantized latent spaces. arXiv preprint arXiv:2211.07292, 2023.

---

### Official Review · Reviewer_SWVW · 2023-10-30

**Soundness:** 2 fair
**Presentation:** 2 fair
**Contribution:** 2 fair
**Rating:** 3
**Confidence:** 4

**Summary:**

The paper proposes SuPr, a novel sub-space prompt learning method to improve the generalization ability of large pre-trained vision language models, especially CLIP. Specially, authors learned several partitions of soft prompts and project them into subspaces while using hard prompts to regularize them. The experiment results show the effectiveness of their method.

**Strengths:**

1.	Improving the generalization ability of pre-trained models is a interesting topic.
2.	Using subspace to enrich the semantic meaning of soft prompts is a interesting direction.

**Weaknesses:**

1.	Results are not consistent. For some dataset, it can achieve slightly better results than SOTA methods, but the results are not good in EuroSAT dataset. The author should explain reasons or assumptions at least.
2.	The experiments are not enough. For example, there is no numerical ablation study for each component.
3.	Overall, the paper is written in a rush way which results in many confusing explanations.

**Questions:**

See the weakness part.

---

> ### Author Response · Authors · 2023-11-21
>
> # Response to SWVW
>
> **W1: Inconsistent Result from EuroSAT dataset.** Please refer to the response to all for this concern.
>
> **W2: Ablation study.** Please refer to the response to all.
>
> **W3: Confusing explanations.** Sorry for providing any confusing explanations. Please kindly let us know which parts confuse you. We will clarify them in a revision.

---

### Official Review · Reviewer_kdeh · 2023-10-31

**Soundness:** 3 good
**Presentation:** 3 good
**Contribution:** 2 fair
**Rating:** 5
**Confidence:** 3

**Summary:**

this paper addresses the prompt learning of vision-language models to achieve better base- and novel-cllass performance with subspace  modelling.  The papers proposes the subspace modelling of soft prompts, as well as its regualization with hard prompts and ensembling methods. Experiments verified the effectiveness of the proposed method.

**Strengths:**

1. the overall method and experiments are reasonable and convincing. This is a good practice for VLMs soft prompting.
2. the paper is well written and easy to follow.
3. the paper marks the first integration of subspace modelling with VLMs.

**Weaknesses:**

the improvement of this paper is not significant according to the Tables (<1% in Table 1, 2,3).

**Questions:**

1. this is a good practice of  integration of subspace modelling with VLMs. How about the novelty of the method in the subspace modellling domain?
3. Why LASP is not compared in Table 3 and Table 4?

---

> ### Author Response · Authors · 2023-11-21
>
> # Response to kdeh
>
> **W1: Performance.** Please refer to the response to all for this concern.
>
> **W2: Novelty in subspace modelling.** Our method SuPr differs from typical subspace modelling methods in the literature by the following aspects: 1) We introduced a novel hard-prompt-based regularization to guide the modelled subspace to span hard-prompt embeddings. This differs from the typical orthogonality regularization commonly used [1-3] to decouple the modelled subspaces of different classes. We also experimented using orthogonality regularization, which induced bad performance. This indicates prompt-based subspace modelling is different from conventional subspace methods. Orthogonalizing them is not beneficial. 2) We proposed an ensembling method to improve our linear subspace modelling by learning multiple linear subspaces with different hard prompts.
>
> [1] You Chong, Daniel Robinson, and René Vidal. Scalable sparse subspace clustering by orthogonal matching pursuit. In CVPR 2016.
>
> [2] Christian Simon, Piotr Koniusz, Richard Nock, and Mehrtash Harandi. Adaptive subspaces for few-shot learning. In CVPR, 2020.
>
> [3] Devos Arnout, and Matthias Grossglauser. Regression networks for meta-Learning few-Shot classification. In AutoML 2020.
>
> **W3: Missing LASP in Table 3&4.** Initially, we noticed a flaw in the LASP paper, as the mean of their reported results did not match their reported mean. Thus, we abandoned their results in Table 3. And, LASP did not provide results for the evaluation for Table 4. Nevertheless, we have now re-implemented LASP for both settings in Table 3/4 and included the results in the revision. The results show that our method consistently holds its superiority over LASP.
>
> Cross dataset transfer (HOS):
> |      | ImageNet（source） | Caltech101 | OxfordPets | StanfordCars | Flowers102 | Food101 | FGVCAircraft | SUN397 | DTD   | EuroSAT | UCF101 | Average |
> | ---- | ------------------ | ---------- | ---------- | ------------ | ---------- | ------- | ------------ | ------ | ----- | ------- | ------ | ------- |
> | SuPr | 71.70              | 94.20      | 89.80      | 64.90        | 70.70      | 86.30   | 23.00        | 66.50  | 45.50 | 50.20   | 67.70  | 65.88   |
> | LASP | 71.40              | 93.30      | 89.88      | 65.01        | 70.20      | 85.39   | 20.88        | 66.74  | 43.67 | 45.32   | 69.07  | 64.95   |
>
> Few shot learning (HOS):
>
> |      | ImageNet | Caltech101 | OxfordPets | StanfordCars | Flowers102 | Food101 | FGVCAircraft | SUN397 | DTD   | EuroSAT | UCF101 | Average |
> | ---- | -------- | ---------- | ---------- | ------------ | ---------- | ------- | ------------ | ------ | ----- | ------- | ------ | ------- |
> | SuPr | 69.77    | 95.17      | 93.13      | 76.80        | 94.23      | 86.00   | 35.53        | 73.60  | 64.97 | 73.23   | 79.97  | 76.58   |
> | LASP | 70.47    | 94.70      | 92.58      | 71.97        | 89.48      | 85.85   | 30.60        | 72.32  | 58.39 | 68.80   | 78.24  | 73.95   |

---

### Official Review · Reviewer_Keaq · 2023-11-01

**Soundness:** 3 good
**Presentation:** 3 good
**Contribution:** 3 good
**Rating:** 6
**Confidence:** 3

**Summary:**

This paper proposes a new subspace-based prompt learning method to search a balance between hand-crafted and learnable prompt. The learn model can achieve high performance on the base classes and it can also generalize to new classes.

**Strengths:**

-The paper is well-written and easy to follow.

-It is interesting to see that the proposed method work well on many datasets.

**Weaknesses:**

-The proposed method fix the parameters of text encoder and image encoder. Will it achieve better performance when making all these parameters learnable.

-Will the proposed training strategy introduce extra training cost?

**Questions:**

See the weakness.

---

> ### Author Response · Authors · 2023-11-21
>
> # Response to Keaq
>
> **W1: Comparison with making all parameters learnable.** We have now included the results of making all parameters learnable during fine-tuning CoOp and SuPr. The results below show that both methods have gained degraded performance when freeing all parameters for training. We attribute this observation to using limited training samples, making over-parameterized models easy to overfit. However, our SuPr still improves over CoOp by about $3.0\$% accuracy in this situation.
>
> | Hos(%)             | ImageNet | Caltech101 | OxfordPets | StanfordCars | Flowers102 | Food101 | FGVCAircraft | SUN397 | DTD   | EuroSAT | UCF101 | Average |
> | ------------------ | -------- | ---------- | ---------- | ------------ | ---------- | ------- | ------------ | ------ | ----- | ------- | ------ | ------- |
> | CoOp               | 71.92    | 93.73      | 94.47      | 68.13        | 74.06      | 85.19   | 28.75        | 72.51  | 54.24 | 68.90   | 67.46  | 71.66   |
> | CoOp w/ all learnable | 59.89    | 92.35      | 91.05      | 65.90        | 62.17      | 80.00   | 23.68        | 71.32  | 55.71 | 68.84   | 76.59  | 69.26   |
> | SuPr               | 73.74    | 96.40      | 96.64      | 75.08        | 85.54      | 91.55   | 37.05        | 80.51  | 70.79 | 81.59   | 81.85  | 79.33   |
> | SuPr w/ all learnable | 62.90    | 93.13      | 91.72      | 67.77        | 68.35      | 82.52   | 28.86        | 73.77  | 59.89 | 76.69   | 78.52  | 72.23   |
>
> **W2: Extra training cost?** **a) Number of trainable parameters:** Our method builds on top of training multiple sets of soft prompts divided from the single set of soft prompts from CoOp, i.e., SuPr has the same size of trainable parameters as CoOp. SuPr-Ens has total parameters = the number of ensembles $\times$ number of parameters of soft prompts, which slightly introduces extra parameters. However, the parameter size of soft prompts is tiny; thus, the extra parameter amount is still small enough. **b) Computational cost:** The following table shows the training time for adapting different models to ImageNet. The results show that our SuPr and SuPr-Ens do not introduce substantially more computational cost  from the baseline ($32$mins of CoOp). CoCoOp requires substantially more cost, $25.5$ times longer than the base unit. The next is ProGrad, which has $2.38$ times the cost of CoOp. Our SuPr and SuPr-Ens scale the training time up to $1.5$ and $1.78$ times of CoOp, which is comparable with the recent SoTA methods, KgCoOp ($1.06$) and LASP ($1.31$).
>
> | GPU: NVIDIA 3090Ti                 | CoOp         | CoCoOp | ProGrad | KgCoOp | LASP | SuPr | SuPr-Ens |
> | ---------------------------------- | ------------ | ------ | ------- | ------ | ---- | ---- | -------- |
> | 10-epoch Training-Time on ImageNet | 1.0 (32mins) | 25.50  | 2.38    | 1.06   | 1.31 | 1.50 | 1.78     |

---

### Author Response · Authors · 2023-11-21

# Response to all

We appreciate the valuable feedback from all the reviewers. We have now addressed all concerns and have uploaded a revised version. In this revision, several new analyses have been included, especially to shed light on the insights of subspace modelling. (Please refer to the 'Qualitative Visualization' in the appendix for more details.)

First, we want to address several important concerns among the reviewers.

**Experimental performance:** Some reviewers raised concerns about the performance of our method, including marginal improvements and underperforming results on EuroSAT.  **a) Marginal improvements:** In the revision, we have added the delta performance between each baseline and our SuPr-Ens in brackets in Table 1. The results show that our method demonstrates superiority over the competitors in the vast majority of cases. Among them, many improvement margins are $>1$% even compared with the latest SoTA methods, KgCoOp and LASP.  **b) EuroSAT results:** We believe the non-ideal EuroSAT results shall be treated as an exception rather than the weakness of our method. LASP shows an extraordinary performance on EuroSAT, outperforming the other SoTA methods, including ProDA, KgCoOp, etc., by over $15\$% accuracy, which should be considered an outlier considering the improvement margins in most prior works. Nevertheless, our SuPr-Ens outperforms all other competitors by more than $7.5\$% accuracy, clearly indicating its efficacy.

**Ablation study:** Apologies for the confusion caused by our ablation experiments shown in Figure 3(b) (and in Table 4 in the appendix) in the submission. We now reiterate and clarify it as follows,

- **CoOp:** Vanilla CoOp, assuming the parameter size of learnable soft prompts is $M$.
- **CoOp-Ensemble:** Multiple CoOp models are introduced, i.e., multiple sets of soft prompts are learnable, whose total parameter size is also $M$. During inference, prediction is based on the ensembling of multiple CoOp models.

- **SuPr w/o reg:** Adding subspace modeling on top of CoOp-Ensemble.
- **SuPr:** Adding hard-prompt based regularization on top of SuPr w/o reg.
- **SuPr-Ens:** Ensembling separate linear subspaces, which are regularized by different subsets of hard prompts for a class.

| Set               | CoOp  | CoOp Ensemble | SuPr w/o reg | SuPr  | SuPr Ens |
| ----------------- | ----- | ------------- | ------------ | ----- | -------- |
| Multiple Prompts  | ✗     | ✔             | ✔            | ✔     | ✔        |
| Subspace Modeling | ✗     | ✗             | ✔            | ✔     | ✔        |
| Regularization    | ✗     | ✗             | ✗            | ✔     | ✔        |
| Subspace Ensemble | ✗     | ✗             | ✗            | ✗     | ✔        |
| Base              | 82.69 | 80.02         | 81.18        | 81.47 | 82.54    |
| New               | 63.22 | 68.51         | 73.30        | 75.21 | 76.36    |
| H                 | 71.66 | 73.82         | 77.04        | 78.21 | 79.33    |

It is seen that CoOp-Ensemble improves CoOp with about $2.16\$% accuracy by learning separately multiple learnable sets of soft prompts and ensembling the learned CoOp models. Incorporating the subspace modelling on top of CoOp-Ensemble to replace the simple ensembling improves its performance further by $3.22\$%, evidencing the effectiveness of SuPr w/o reg. Furthermore, adding the hard-prompt-based regularization pushes the model performance up by $1.17\$%, as the achievement of SuPr, which is further improved with $1.12\$% accuracy by subspace ensembling -- SuPr-Ens.

---

### Author Response · Authors · 2023-11-23
**Feedback on our rebuttal.**

Dear reviewers,

Please have a look at our rebuttal and let us know if there are any further revisions or adjustments you would like us to make.
Alternatively, please kindly consider raising your scores if there are no additional concerns.

Best regards,

The Authors

---

### Meta-Review · Area_Chair_VA2v · 2023-12-02

**Metareview:**

The authors propose a novel subspace-based prompt learning method, named SuPr, which can effectively model subspaces spanning the embeddings of both the learnable soft and the textual/hard prompts. Hand-crafted prompts are further used to regularize our subspace-based alignment between hand-crafted prompts and learnable prompts to achieve excellent fitting of base classes and generalization to novel classes. The proposed method is evaluated on 11 diverse image classification datasets.
Pros:
* The method is simple and efficient.
* Many experimental results.
Cons:
* Only marginal improvement.
* Improvement is not not consistent (e.g., EuroSAT).

The authors tried to address reviewers' concerns very proactively.
Unfortunately, only SWVW responded to authors rebuttal.
SWVW disagrees that EuroSAT is only an outlier. The updated Table 1 also clearly shows that the improvement is not consistent.
The AC agrees with SWVW's judgment and recommends rejection.

**Justification For Why Not Higher Score:**

The rebuttal did not address the concerns of two weakness below.
* only marginal improvement.
* improvement is not not consistent (e.g., EuroSAT).

**Justification For Why Not Lower Score:**

N/A

---

### Decision · Program_Chairs · 2024-01-16

Reject